# The Research of Interface Microdomain and Corona-Resistance Characteristics of Micro-Nano-ZnO/LDPE

**DOI:** 10.3390/polym12030563

**Published:** 2020-03-04

**Authors:** Yujia Cheng, Guang Yu

**Affiliations:** Mechanical and Electrical Engineering Institute, University of Electronic Science and Technology of China, Zhongshan Institute, Zhongshan 528400, China; chengyujia@hrbust.edu.cn

**Keywords:** LDPE, Micro, Nano ZnO, interface microdomain, crystalline morphology, corona-resistant properties

## Abstract

In this article, the melting blend was used to prepare the Micro-ZnO/LDPE, Nano-ZnO/LDPE and Micro-Nano-ZnO/LDPE with different inorganic particles contents. The effect of Micro-ZnO and Nano-ZnO particles doping on interface microdomain and corona-resistance breakdown characteristics of LDPE composite could be explored. Based on the energy transfer and heat exchange theory of energetic electrons, the inner electrons energy transfer model of different ZnO/LDPE composites was built. Besides, the microstructure and crystalline morphology of inorganic ZnO-particles and polymer composites were detected by SEM, XRD, FTIR, PLM and DSC test, and the AC breakdown and corona-resistance breakdown characteristics of composites could be explored. From the experimental results, the Nano-ZnO particles after surface modification dispersed uniformly in LDPE matrix, and the nanoparticles agglomeration almost disappeared. The inorganic particles doping acted as the heterogeneous nucleation agent, which improved the crystallization rate and crystallinity of polymer composites effectively. The ZnO particles with different size doping constituted the different interface structure and crystalline morphology, which made different influence on composites macroscopic properties. When the Nano-ZnO particle size was 40nm and the mass fraction was 3%, the breakdown field strength of Nano-ZnO/LPDE was the highest and 15.8% higher than which of pure LDPE. At the same time, the shape parameter *β* of Micro-Nano-composite was the largest. It illustrated the microparticles doping reduced the probability of nanoparticles agglomeration in matrix. Besides, both Micro-ZnO and Micro-Nano-ZnO particles doping could improve the ability of corona corrosion resistance of LDPE in varying degrees. The corona-resistant breakdown time order of four samples was as follows: LDPE < Micro-ZnO/LDPE < Nano-ZnO/LDPE < Micro-Nano-ZnO/LDPE. When the mass fraction of Micro-ZnO and Nano-ZnO particles was 2% and 3% respectively, the corrosion depth and area of Micro-Nano-ZnO/LDPE was the least, and the ability of corona corrosion resistance was the strongest.

## 1. Introduction

As the modern power system developed, the higher requirements on load and stress level of electrical insulation was put forward. Therefore, the insulation materials with excellent properties became a research focus in the electrical area. With the excellent electrical, mechanical, physical and chemical properties, polythene was widely used in electrical equipment insulation. However, under high electric stress for a long time, the partial discharge generated in polythene, which initiated the growing of electrical trees. Besides, the cables’ operation life reduced, which affected the safe working of the power system. Therefore, how to improve the long-term dielectric properties of polythene insulation has become urgent to solve [1,2,3,4]. Research showed that the conventional polymer/inorganic Micro-composites possessed excellent thermal conductivity and thermal blocking effect. Besides, the effect of electrical corrosion resistance was obvious. For the nanoparticles doping, it could improve many electrical properties of polymers such as partial discharge resistance, electrical treeing resistance, corona aging resistance and space charge accumulation restraint [5,6,7]. Therefore, a new composite material combining both of these advantages was our aim to find out.

On this account, the Micro-Nano-composite technology was used to regulate and control the interface bonding state between the polymer and inorganic Micro-Nano-particles. Then the mesoscopic morphology of composites was regulated and controlled. For one thing, the inorganic Nanophase would form the quasi-homogeneous phase structure in polymer matrix, for another it could absorb and consume the energy of energetic particles produced by discharge according to the interface effect of inorganic microparticles and nanoparticles. Fundamentally, it could improve the long-term dielectric properties and operation life. The interface microdomain was one of the most important factors which effected the microstructure and macroscopic properties of Micro-Nano-composites. The formation of interface microdomain not only depended on the property, scale and shape of inorganic filler and matrix resins, but also closely related to the composite technology, the kind of compatibilizer and sample preparation methods [8,9,10,11]. Although the dielectric properties of Micro-Nano-composites were widely concerned, the major research works were focused on epoxy resin composites and silicon rubber composites; the other engineering dielectric materials needed to be further explored. Besides, most of the content in references were the conclusions related to the test results of composites, while the necessary analysis about correlation between dielectric properties and composite microstructure were omitted. Especially the research of interface morphology, breakdown characteristics and corona-resistant corrosion properties of complex composites with microparticles and nanoparticles doping simultaneously were a lack of research. In this article, according to the research on interface microdomain conformation of Micro-Nano-composites, the mechanism of composite mesostructure on Micro-Nano-composite dielectric behavior could be explored. Besides, the Micro-ZnO and Nano-ZnO particles was doped into the LDPE matrix. According to the corona-resistance breakdown test combining with some microstructure characterization techniques such as SEM, XRD, FTIR, PLM and DSC, the relationship between interface microdomain morphology and properties of polymer/inorganic Micro-Nano-composites was built. At the same time, the effect of different sized ZnO particles doping on dielectric strength of LDPE was explored.

## 2. Experimental Method

### 2.1. Organic Treatment of ZnO Particles

In this article, the ZnO particles were dealt with surface modification by silane coupling agent KH-570. Under certain conditions, this coupling agent was adsorbed to inorganic surface by physical or chemical reaction [12,13]. According to the dehydration condensation reaction, the silane coupling agent reacted with the hydroxyl on ZnO surface, from which the stronger bonding between the molecules would be formed. After being heated or dried, the covalent bond was formed according to the dehydration condensation reaction. Besides, the coupling agent had good affinity with LDPE. According to the organic treatment, the ZnO surface diverted from hydrophilic to lipophilic, from which the ZnO particles could be combined with LDPE better and the perfect interface structure was formed [14,15]. The surface modification mechanism of silane coupling agent is shown in Figure 1.

### 2.2. Preparation of Micro-Nano-ZnO/LDPE

The raw materials information: Both Nano-ZnO and Micro-ZnO particles were spherical structure, of which particle size were 40nm and 1µm respectively. The ZnO particles were purchased from Beijing Deke Daojin Science and Technology Co., Ltd. (Beijing, China). The silane coupling agent (KH-570) was produced by Shanghai Energy Chemical. Low density polyethylene (LDPE) was produced by Sinopec Group.

With the torque rheometer RM-200A, the Micro-ZnO/LDPE, Nano-ZnO/LDPE and Micro-Nano-ZnO/LDPE was prepared by double melt blending [16]. Firstly, the LDPE, Nano-ZnO particles and Micro-ZnO particles, after surface modification, were added into the Xylene solution together. At 80 °C, the mixture was dealt with and electric stirred for 2 h. Then, these samples were dried and the masterbatch was obtained. At last, the masterbatch and LDPE was added into the torque rheometer in the certain proportion. At 150 °C, the Micro-ZnO/LDPE, Nano-ZnO/LDPE and Micro-Nano-ZnO/LDPE was prepared by melt blending for 20min. Among them, the mass fraction of inorganic particles in both Micro-ZnO/LDPE and Nano-ZnO/LDPE were 1%, 3% and 5% respectively; while the mass fraction of Micro-ZnO and Nano-ZnO in different Micro-Nano-ZnO/LDPE were 1% and 3%, 2% and 3%, 3% and 3% respectively. In this article, these samples were marked with M1, M3, M5, N1, N3, N5, N3M1, N3M2 and N3M3 respectively.

### 2.3. Structure Characterization of Different Samples

In this article, the SEM(S-4800) produced by Hitachi was used to observe the microparticles and nanoparticles. During the SEM test, a small amount of ZnO powder was applied to conductive adhesive and observed by SEM, from which the size and shape of ZnO particles could be explored.

The X-ray diffraction D8-FOCUS was produced by Bruker. The Cu target was used for scanning, of which the range was between 1.5° and 90°. The scan rate was 0.06°/s and the wavelength *λ* was 0.15406 nm. The tube voltage and experiment’s current was 40 kV and 30 mA respectively. Under the test conditions above, the nanoparticle structure of composites was explored. Besides, the molecule structure and morphology within the samples was measured.

The different samples were tested by FTIR (EQUINOX55). During the test, the KBr was used for background, and the wavelength range was 4000~500 cm^−1^. Besides, the sample’s thickness was 100 μm.

The crystalline morphology and crystal size of Micro-ZnO/LDPE, Nano-ZnO/LDPE and Micro-Nano-ZnO/LDPE was tested by PLM. Firstly, the samples were dealt with surface corrosion by Concentrated Sulphuric Acid and Potassium Permanganate with 5% contents. After the ultrasonic cleaning, these samples were placed on glass slides. Then the crystalline morphology of ZnO/LDPE composites could be observed by polarization microscope.

To further explore the crystallization process of pure LDPE, Micro-ZnO/LDPE and Nano-ZnO/LDPE, the parameter changes of crystallization process was tested by DSC-1, which was produced by the Mettler Toledo Company. This experiment was carried out under nitrogen protection, of which flow rate was 150 mL/min. Both heating and cooling rate was 10 °C/min. According to the melting enthalpy ΔHm, the samples crystallinity could be calculated by Formula (1).
(1)$$X_{c}=\frac{\Delta H_{m}}{\left(1-\omega \right)H_{0}}\times 100\%$$

In Formula (1), Δ*H_m_* was the melting enthalpy; *H*_0_ was the material melting enthalpy in fully crystallization, of which PE was 293.6 J/g (calibration in experimental instrument); *ω* was the ZnO mass fraction of composites [17].

### 2.4. Breakdown Test of Different Samples

In this article, the AC power frequency system was used for breakdown test. This system boosted at a speed of 1 kV/s until the material breakdown, and the breakdown field strength *U* was recorded. Then the thickness of breakdown points *d* was measured. According to the formula *E* = *U*/*d*, the breakdown field strength of different samples could be calculated. The experimental device is shown in Figure 2.

The thickness of test samples was 100 μm. Each sample was tested for 30 breakdown points. In order to eliminate the material history effect, these samples must be dealt with pretreatment before breakdown test [18]. Then the tested samples were placed into the vacuum drying oven of which temperature was 80 °C for 24 h. In case the surface discharge happened in these samples, the whole electrode system and samples were dipped into the cable oil together during the test. The transformer boosted at a uniform speed until the short circuit alarm was sent out. Stopping pressure and the applied voltage at this time was the samples’ breakdown voltage. These data were recorded and the thickness of breakdown points *d* was measured, from which the breakdown field strength of composites could be calculated. At last, the MINITAB was used to analyze the data and the Weibull distribution curve was drawn. The Weibull distribution expression of the two most important parameters is shown in Formula (2).
(2)$$P\left(E\right)=1-\exp(-\left(\frac{E}{E_{0}}{)}^{\beta }\right)$$

From Formula (2), *P* was the breakdown probability of samples. *E* was the breakdown field strength of samples. *β* was the shape parameter which characterized the data dispersion. *E*_0_ reflected the scale parameter of breakdown field strength when *P* was 63.2%.

By calculation, Formula (2) could be changed to Formula (3).
(3)$$\ln(-\ln(1-p))=\beta \left(\ln E-\ln E_{0}\right)$$

According to Formula (3), *E*_0_ could be calculated. The breakdown probability of each sample could be calculated by Formula (4).
(4)$$P_{i}=\frac{i-0.44}{n+0.25}\times 100\%$$

In Formula (4), *i* was the breakdown field strength test times in ascending sort order of each sample. *n* was the total test times of each sample.

### 2.5. Corona-Resistance Breakdown Test of Different Samples

The needle-plane electrode system was used for corona breakdown test [19]. The top electrode needle was made by pure tungsten, of which curvature radius was 100 ± 2 μm. The lower electrode plane was made by stainless steel. The clearance between electrodes was 2 mm. In corona discharge experiment, the applied voltage of needle-plane electrode was 4 kV. The corona aging experimental device was made up of voltage source, voltage regulator, step-up transformer, conductive paper, protective resistance and pH Electrode holder. Among them, the protective resistance was used to prevent the transformer or voltage regulator from being damaged by high current which generated in the moment of samples’ breakdown. The conductive paper was the experimental protection device, which was connected with tungsten needle electrode. When the high current existed in circuit, the conductive paper heated up and burnt instantly. Then, the circuit was cut off, which could protect the experimental apparatus. The detecting electrode system is shown in Figure 3.

In this experiment, the needle electrode system did not contact with the sample’s surface directly. According to the corona discharge, the electricity was spread by needle tip in arcuate form. Firstly, eight samples per composite were dealt with corona discharge corrosion until these samples were breakdown. Then, the breakdown time of each sample was recorded and the average breakdown time was calculated, from which the ability to corona-resistance breakdown of different composites could be explored. At last, according to the result of corona-resistance breakdown test, the influence mechanism of Micro-ZnO and Nano-ZnO particles to breakdown property of polyethylene matrix was explored. In this experiment, the industrial frequency AC power was used for the test.

## 3. Experimental Result and Analysis

### 3.1. Morphology Characterization of ZnO Particles with Different Size

The SEM test results of microparticles and nanoparticles are shown in Figure 4, from which the size and shape of inorganic ZnO particles could be observed clearly.

From Figure 4, when the Nano-ZnO particle size was 40 nm, all the particles were basically the same. Besides, the distribution of particle size was smaller, and the particles were near-spherical shape. When the Micro-ZnO particle size was 1 µm, most of the particles’ size were larger, and the distribution of particles’ size increased evidently. However, this time, the particles’ shapes were more irregular. Most microparticles were near-spherical shape, but a few particles were rod shape.

### 3.2. XRD Characterization of Composites

The X-ray diffraction (XRD) was the characterization of crystal geometry configuration, which was a commonly used method to detect the crystalline morphology of polymer material. It could identify the size of interlayer space and structure parameters of Nanoparticles. According to the diffraction angle corresponding to X-ray diffraction peak, the crystal lattice periodic array spacing of inorganic crystals was calculated. Besides, the changes caused by the structural interaction of nanoparticles and polymers was explored [20]. In this experiment, the Nano-ZnO particles and Nano-ZnO/LDPE before and after modification was detected. The Nano-ZnO particle size was 40 nm. The XRD test results are shown in Figure 5.

From XRD patterns, the position of Nano-ZnO diffraction peak before and after modification was unchanged. The diffraction peak existed in 2θ = 31.884°, 34.72°, 36.571°, 47.837°, 56.936°, 63.121° and 68.202° respectively. Compared with the standard card JCPDS72239 [21], it could be concluded that the Nano-ZnO still related to hexagonal crystal structure after modification. On the other hand, after modification, the diffraction peak narrowed and diffraction intensity increased. This was because the bonding interaction was formed between Nano-ZnO particles and silane coupling agent, which increased the diffraction intensity of nanoparticles. From XRD patterns of Nano-ZnO/LDPE, the broad peak appeared at lower angles of ZnO/LDPE samples. On the corresponding position of LDPE (110) plane and (200) plane, that 2θ = 22.154° and 2θ = 24.478°, the diffraction peaks appeared. Besides, the Nano-ZnO diffraction peak was hardly to be found in XRD patterns of Nano-ZnO/LDPE. This was because the Nano-ZnO particles’ reunion scale and probability decreased in polyethylene matrix, and the nanoparticles’ dispersion was uniform [22].

### 3.3. FTIR Characterization of Different Samples

According to the FTIR test, the characteristic absorption peak of different samples and ZnO particles before and after modification were analyzed qualitatively. The polymer structure and functional groups’ quantity were judged by different absorption peak position and intensity. Besides, the interface interaction between ZnO particles and LDPE matrix was explored. The FTIR patterns of different samples and ZnO particles before and after modification are shown in Figure 6.

From Figure 6a, the N–H absorption peak appeared around 3085 cm^−1^, the C–H weak stretching vibration peaks appeared around 2925 and 2846 cm^−1^, the Si–O–C characteristic peak appeared around 1075 cm^−1^ of organic ZnO. Besides, the –CH2CH2CH2– absorption peak, the –CH2– out-of-plane vibration absorption peak and –CH2CH2CH2– deformation vibration peak appeared around 734 cm^−1^, 603 cm^−1^ and 488 cm^−1^ respectively. All these indicated that the silane coupling agent was grafted to the Nano-ZnO particles’ surface. The Nano-ZnO particles were coated with silane coupling agent, from which the Nano-ZnO particles possessed the lipophilic group and combined with polymer matrix better. The nanoparticles would disperse uniformly in polymer. From Figure 6b, there were three high intensity characteristic absorption peaks around 723, 1450 and 2850 cm^−1^. They were C-H rocking vibration peak, C-H bending vibration peak and C-H symmetrical and asymmetrical stretching vibration peak respectively. At the same time, there were two split peaks around 2850 cm^−1^. This indicated that the ZnO particles doping did not change the characteristic peak position in LDPE matrix, that was, the ZnO particles did not change the molecular chain structure of LDPE. Around 3500 cm^−1^, the hydroxyl vibration peak of microparticles and nanoparticles almost disappeared because the microparticles and nanoparticles were coated by LDPE macromolecules and were hard to be detected in composites’ preparation. Besides, the –OH on ZnO particles’ surface would react with –H to form the water molecules during the composites’ preparation. Then, the water molecules would evaporate from materials. From the FTIR patterns of composites, the Zn–O–Si vibration absorption peak appeared around 1200 cm^−1^. This indicated that the silane coupling agent reacted with the hydroxyl on ZnO particles’ surface, and these particles were coated by silane coupling. The surface of inorganic ZnO particles were provided with lipophilic groups, which improved the compatibility of ZnO particles and matrix resin.

### 3.4. DSC Characterization of Different Samples

The isothermal crystallization and melting process parameters of Micro-ZnO/LDPE, Nano-ZnO/LDPE and Micro-Nano-ZnO/LDPE with different mass fraction of ZnO particles are shown in Table 1.

According to the experimental results of DSC, the crystallization peak temperature *T_c_*, the melting temperature *T_m_* and the crystallinity *X_c_* of these composites were all higher than that of pure LDPE. The Micro-ZnO and Nano-ZnO particles doping played the role of nucleating agent, which advanced the ordered arrangement of LDPE molecular chain. The macromolecule chains of pure LDPE existed around the ZnO particles. When the mass fraction of different ZnO particles were all 5%, the crystallinity order of four samples was as follows: LDPE < M5 < N5 < N3M2. When the number of ZnO particles increased, the proportion of crystallization per volume in polymer grew. Therefore, the crystallinity of all ZnO/LDPE composites was higher than pure LDPE. With the same mass fraction of different ZnO particles doping, the number of inorganic particles in Nano-ZnO/LDPE was the largest. While the mass fraction of nanoparticles was 5%, the particles’ reunion existed in Nano-ZnO/LDPE. Therefore, the crystallinity of Nano-ZnO/LDPE was lower than Micro-Nano-ZnO/LDPE but higher than Micro-ZnO/LDPE. Besides, as the inorganic materials, the ZnO particles provided excellent thermal conductivity and heat resistance, from which the heat would be transferred quickly. Therefore, the melting temperature *T_m_* of ZnO/LDPE composites was higher.

### 3.5. Breakdown Test of Different Samples

In order to explore the effect of different Nano-ZnO particle mass fraction on composites’ breakdown field strength, the samples with 1, 3 and 5 wt% mass fractions were chosen for AC breakdown test. The test data were analyzed by Weibull distribution, which are shown in Figure 7.

From Figure 7, the breakdown field strength of different composites were all higher than that of pure LDPE. With the increasing of Nano-ZnO mass fraction, the breakdown field strength of Nano-composites first increased and then decreased. When the mass fraction was 3 wt%, the breakdown field strength of Nano-composite was the highest and 15.8% higher than that of pure LDPE. This was because the nanoparticles doping changed the internal structure of polythene matrix. When the particles’ mass fraction was lower, the nanoparticles dispersed uniformly in polymer matrix under effect of applied electric field. The distance between the electric double layer formed by nanoparticles was greater. Besides, the nanoparticles’ quantity was less, and the contact area with polymer matrix was smaller. Therefore, the traps’ quantity was less and the electrons were hard to be captured, but the breakdown strength increased. With the increasing of mass fraction, the distance between the electric double layer formed by nanoparticles decreased, and the interaction regions drew toward each other. Besides, the nanoparticles’ quantity increased, and the contact area with polymer matrix was larger. The internal structure of LDPE matrix was affected greatly. Therefore, the traps’ quantity increased and the electrons were easy to be captured. The energy barrier produced by electron transition increased, and the material breakdown strength was the highest [23,24]. When the mass fraction was 5 wt%, the nanoparticles’ quantity was the largest. The distance between the electric double layer was closer, and the interaction regions interacted with each other. The traps’ quantity increased further, and the electrons were captured more easily, but in the electric double layer, there was a conducting path which was produced by free electrons. The carrier was easy to migrate, which decreased the electronic transitions barrier. So, the breakdown field strength of Nano-composite decreased.

The breakdown field strength test result of different Micro-composites with particles’ mass fraction of 1%, 3% and 5% are shown in Figure 8.

From Figure 8, with the increasing of microparticles’ mass fraction, the breakdown field strength of composites decreased gradually. This was because the microparticles did not possess some special effects of nanoparticles, such as small size and large specific surface area. Besides, only weak intermolecular force existed between microparticles and polymer matrix. The microparticles were equivalent to an impurity, which blocked the chain segment structure of polymer macromolecule chains. These flaws were fatal in the process of electric field explosion. Therefore, the breakdown field strength of Micro-ZnO/LDPE composites decreased.

From the breakdown test result of Nano-composites, when the nanoparticles’ mass fraction was 3 wt%, the breakdown field strength of Nano-composites was the highest. In order to explore the Micro-Nano-synergetic effect on breakdown characteristics of Micro-Nano-ZnO/LDPE, the Nano-ZnO/LDPE with 3% nanoparticles’ mass fraction was used for matrix. Then the microparticles were added into the matrix until the microparticles’ mass fraction were 1%, 2% and 3% respectively, and the Micro-Nano-composites N3M1, N3M2 and N3M3 were prepared. These samples were dealt with AC voltage withstand test. At last, the breakdown field strength of three Micro-Nano-composites were obtained in Weibull distribution, which are shown in Figure 9.

From Figure 9, in same mass fraction, the breakdown field strength of Micro-Nano-composites were higher than that of pure LDPE. In three Micro-Nano-composites, the breakdown field strength of sample N3M2 was the highest. It was 124.6 kV/mm and 3.2% higher than that of pure LDPE, but the breakdown field strength of Nano-composites were higher. This was because the microparticles did not possess the characteristics as well as nanoparticles. When the microparticles were added into the Nano-composites, the microparticles reacted with polymer matrix, which destroyed the original structure of polymer matrix. At the same time, the microparticles doping increased the nucleating agent quantity in polymer. The non-crystalline regions decreased. As the shallow traps were produced by LDPE molecule chains buckling and breakage in non-crystalline regions, the reduction of non-crystalline regions decreased the shallow trap quantity. Besides, the microparticles doping would destroy the interface structure between nanoparticles and polymer matrix. The deep traps’ quantity decreased, which affected the breakdown field strength of Micro-Nano-composites. Furthermore, the microparticles possessed excellent mechanical and thermal conductivity, which decreased the heat accumulation of Micro-Nano-composite. The reduction of thermionic quantity decreased the probability of polymer matrix destruction. The shape parameter *β* of Micro-Nano-composite was larger than that of Nano-composite. It illustrated that the inorganic particles dispersed uniformly in polythene matrix due to the microparticles doping, which decreased the nanoparticles agglomeration.

### 3.6. Corona-Resistant Breakdown Test of Different Samples

In order to further discuss the effect of Micro-ZnO and Nano-ZnO particles doping to dielectric properties of LDPE, after the pretreatment, the samples LDPE, N5, N3M2 and M5 were placed in needle-plane electrode system for corona-resistant breakdown test. The experimental results are shown in Figure 10.

From Figure 10, the corona-resistant breakdown time of N5, N3M2 and M5 were all longer than that of pure LDPE, which illustrated the different ZnO particles doping improved the ability of corona corrosion resistance of pure LDPE in varying degrees. The corona-resistant breakdown time order of four samples was as follows: LDPE < M5 < N2 < N3M2.

### 3.7. PLM Characterization of Different Samples

The PLM was used to test different samples after corrosion for 16h, and the effect of different ZnO particle size on the corona-resistant properties of polymer could be further observed. The corrosion results of four test samples is shown in Figure 11.

From Figure 11, in same corrosion time, the corrosion degree and morphology of four samples was different. Among them, the ability of corona corrosion resistance of different ZnO/LDPE composites was all stronger than that of pure LDPE. From Figure 11a, after corona corrosion, the larger gully appeared on samples’ surface. According to the microscope focus, the corroded depth all around the pure LDPE surface was different, and the samples’ surface was severely damaged. From Figure 11b, no obvious corrosion damaged areas were found on the surface of Nano-composites, but parts of crystalline areas appeared faintly. From Figure 11c, the corona corrosion damaged areas also appeared in Micro-composite, but the corrosion degree was less than that of pure LDPE. From Figure 11d, the marks of corona corrosion could be found in the surface of Micro-Nano-composite, but the damaged area was very small.

### 3.8. Exchange Mechanism of Electron Energy

Based on the intrinsic electric breakdown theory of solid dielectric, the electrons obtained the energy from strong electric field, from which the electrons would be accelerated to turn into thermionic. In the process of directional migration, the thermionic would interact with dielectric, from which the electric field energy was transferred to dielectric directly. When the electron energy which was obtained from electric field was stronger than polymer bond energy, the polymer molecular chains would break, and the melting produced by local high temperature would appear. In corona-resistant test, the local high electric field produced by needle electrode caused the electron emission or gas ionization around needle tip, which would produce a large number of energetic electrons. According to the electrons’ bombardment, the electric corrosion effect was formed in samples’ surface. For the polymer with inorganic particles doping, the inorganic particles played the role of excluding and dissipating heat in the process of corrosion, which would reduce the damage of polymer segments. From the test result of corona-resistant property of different composites, the structure of Micro-Nano-ZnO/LDPE was close, and the ability of corona corrosion resistance was the strongest. The corona corrosion resistant process of polymer composites is shown in Figure 12.

Combined with the results of corona-resistant breakdown test and the process of composites’ corona corrosion resistant, the amorphous region between ZnO particles was corroded by electric field in corona aging experiment, from which the Micro-ZnO and Nano-ZnO particles exposed gradually because the ZnO particles possessed high dielectric constant and excellent thermal conductivity, which could limit the corona corrosion. Besides, in process of corona aging, the ZnO particles could reduce the damage of heat to the composites [25]. From the experimental result of DSC, the ZnO/LDPE composites possessed higher crystallinity, and the grains’ arrangement was close, which prolonged the samples’ breakdown time. In conclusion, the inorganic particles doping could improve the corona-resistant properties of polymer.

Based on the energy transfer and heat exchange theory of energetic electrons, the inner electrons energy transfer model of different ZnO/LDPE composites was built as shown in Figure 13.

With the same mass fraction of inorganic ZnO, the number of Micro-ZnO particles was less, and the area of non-crystalline region was larger. Therefore, the local deep corrosion would appear in exposure area of polymer. For the Micro-ZnO/LDPE, it possessed a large geometrical size and better thermal conductivity. When the electrons bombarded on the Micro-ZnO particles, the electrons’ migration path would change due to the particles’ blocking effect, which caused the energy consumption. On the other hand, according to the particles’ heat conduction, the local heat, which was produced by electron high energy bombardment, was dissipation. Therefore, the damage of polymer molecular chains reduced. For the Nano-ZnO/LDPE, owing to the small size of Nano-ZnO particles, these inorganic particles would be dispersed uniformly in polymer. Therefore, the area of non-crystalline region which was exposed to the situation of corona discharge was less. Under the effect of electron bombardment, the area of electric corrosion was less, but owing to the small size of nanoparticles, the ability of thermal dissipation was weaker than that of microparticles. Therefore, the heat produced by electron bombardment would cause the Nano-ZnO particles to sink with the polymer melting, and the whole thickness of Nano-ZnO/LDPE samples was thinned. For Micro-Nano-ZnO/LDPE, the quasi-average distribution was formed by nanoparticles and microparticles [26]. According to the blocking scattering effect of nanoparticles and the thermal dissipation of microparticles, the effect of electrical corrosion resistance of Micro-Nano-ZnO/LDPE was the most obvious.

## 4. Conclusions

In this article, the Micro-ZnO and Nano-ZnO particles with different particle size was doped into pure LDPE. The Micro-ZnO/LDPE, Nano-ZnO/LDPE and Micro-Nano-ZnO/LDPE were prepared by double melting blend. The Nano-ZnO particles were dealt with surface modification by silane coupling agent. Then, the inorganic ZnO particles and composites were dealt with crystalline morphology test by SEM, XRD, FTIR, DSC and PLM. From the result of AC breakdown test and corona-resistant breakdown test, the synergistic effect between microparticles and nanoparticles on dielectric properties of polymer was explored. The conclusions were as follows:

(1) From the test result of XRD: After the surface modification by silane coupling agent, the crystalline structure of ZnO did not change. However, for Nano-ZnO particles after surface modification, its diffraction peak narrowed and the diffraction intensity increased. It illustrated that a strong bonding interaction was formed between Nano-ZnO particles and silane coupling agent. Besides, the Nano-ZnO diffraction peak was hardly to be found in XRD patterns of Nano-ZnO/LDPE. Combined with the results of FTIR test, the Nano-ZnO particles’ reunion scale and probability decreased in polyethylene matrix, and the nanoparticles’ dispersion was uniform.

(2) From the test result of DSC: Both Micro-ZnO and Nano-ZnO particles doping could improve the crystallinity and crystallization rate of pure LDPE effectively, which changed the crystalline structure of polymer. Then the different interface structure was formed. Among them, the crystallinity of Micro-Nano/LDPE was the highest.

(3) From the result of AC breakdown test: The breakdown field strength of Nano-ZnO/LDPE was the highest. When the mass fraction of Nano-ZnO particles was 3%, the breakdown field strength of Nano-ZnO/LDPE was 15.8% higher than that of pure LDPE. The breakdown field strength of Micro-Nano-ZnO/LDPE and Micro-ZnO/LDPE were the middle and the lowest respectively. But the Weibull shape parameters of Micro-Nano-ZnO/LDPE was larger. Therefore, the microparticles doping was helpful for the nanoparticle dispersion in matrix, from which the Micro-Nano-synergetic effect worked more effectively. According to the result of corona-resistant breakdown test, both Nano-ZnO and Micro-ZnO particles doping could improve the ability to corona-resistance breakdown of pure LDPE. The ability of corona-resistant order of four samples was as follows: LDPE < Micro-ZnO/LDPE < Nano-ZnO/LDPE < Micro-Nano-ZnO/LDPE. Among them, the ability of corona-resistance of Nano-ZnO/LDPE and Micro-ZnO/LDPE was 2 times and 1.5 times higher than that of pure LDPE.

(4) Combining the PLM test result of four samples after corrosion for 16h, the inhibition effect of different sized ZnO particles doping on corona-resistant property of polythene could be verified. Based on the energy transfer of energetic electrons, the inner electrons energy transfer model of different ZnO/LDPE composites was built. Besides, according to the corona corrosion-resistant property pattern of inorganic particles/polyethylene composites, the influence mechanism of Micro-ZnO and Nano-ZnO particles to corona-resistant property of polymer could be further explored.

## Figures and Tables

**Figure 1 polymers-12-00563-f001:**
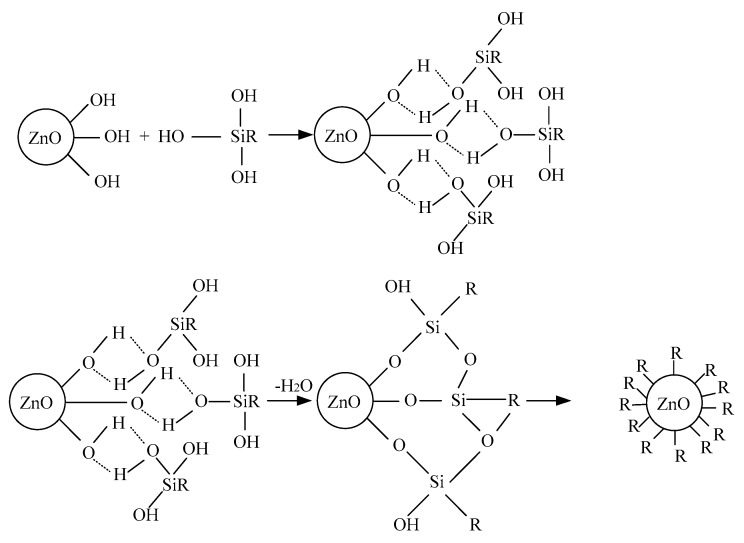
Surface modification of ZnO with silicane coupling agent (KH570).

**Figure 2 polymers-12-00563-f002:**
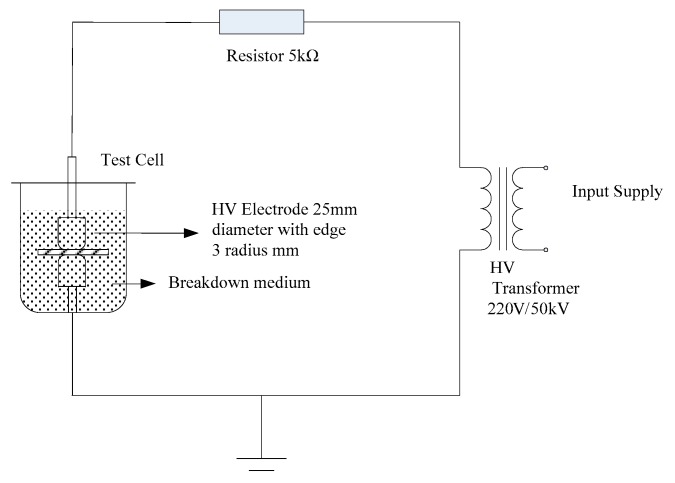
Experimental device for the AC breakdown test.

**Figure 3 polymers-12-00563-f003:**
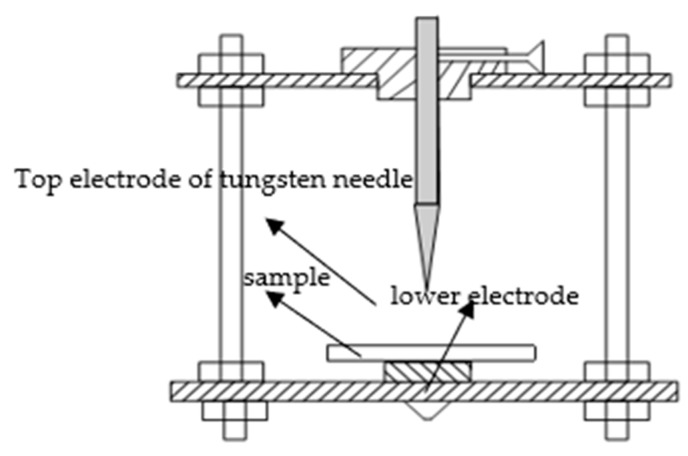
Electrode system for the corona breakdown test.

**Figure 4 polymers-12-00563-f004:**
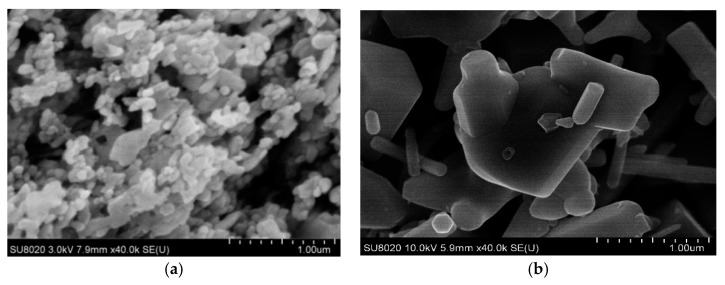
Morphology characterization of (**a**) Nano-ZnO (40 nm); (**b**) Micro-ZnO (1 µm).

**Figure 5 polymers-12-00563-f005:**
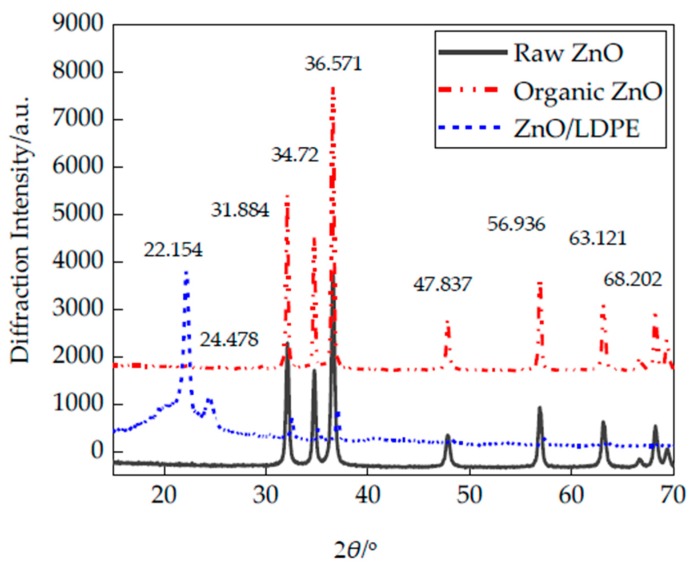
XRD patterns of ZnO particles and ZnO/LDPE specimens.

**Figure 6 polymers-12-00563-f006:**
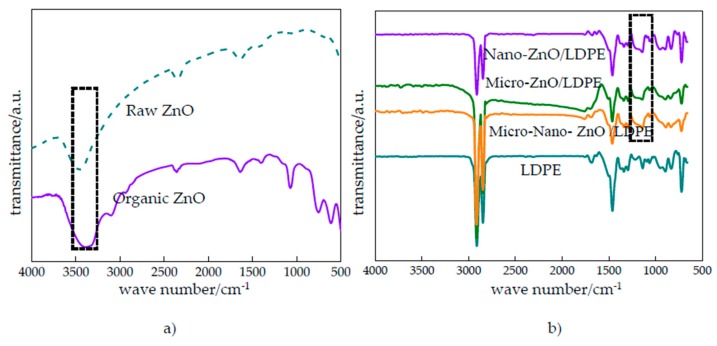
FTIR patterns of (**a**) ZnO particles; (**b**) LDPE, Nano-ZnO/LDPE, Micro-ZnO/LDPE and Micro-Nano-ZnO/LDPE specimens.

**Figure 7 polymers-12-00563-f007:**
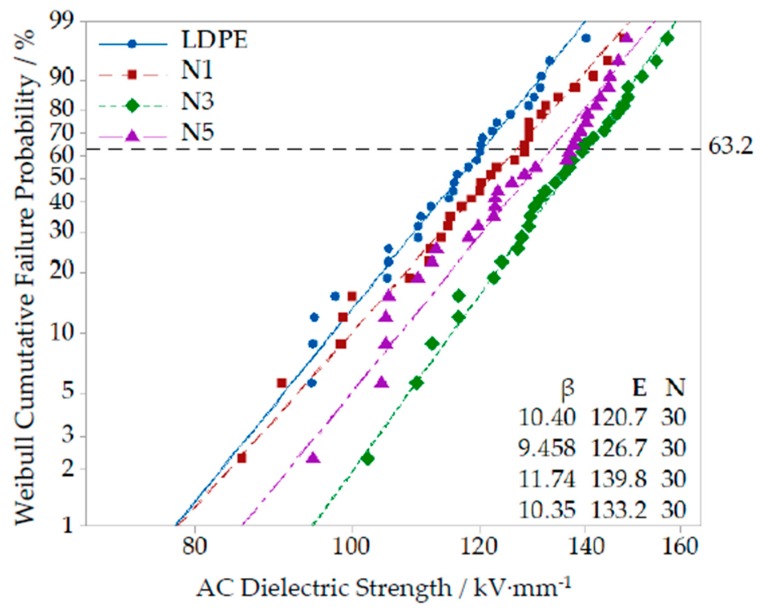
AC breakdown field strength Weibull diagram of Nano-composites and pure LDPE.

**Figure 8 polymers-12-00563-f008:**
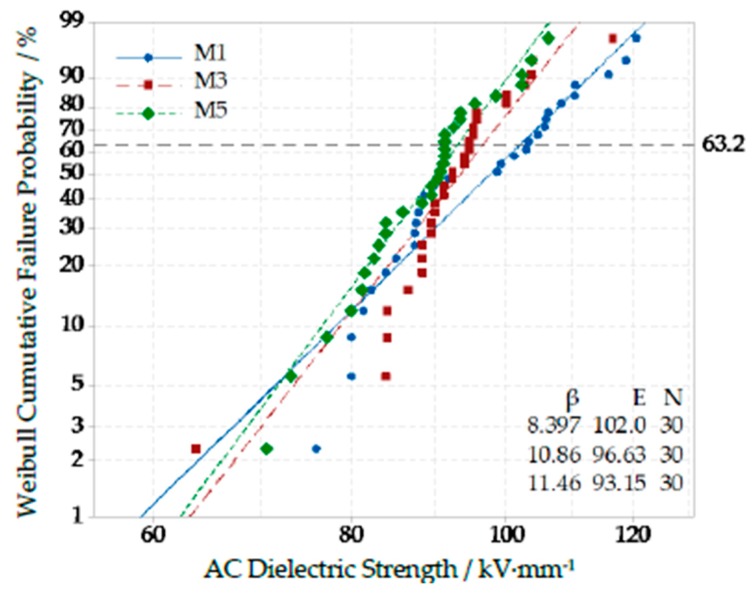
Weibull distribution curve of different Micro-ZnO content composites.

**Figure 9 polymers-12-00563-f009:**
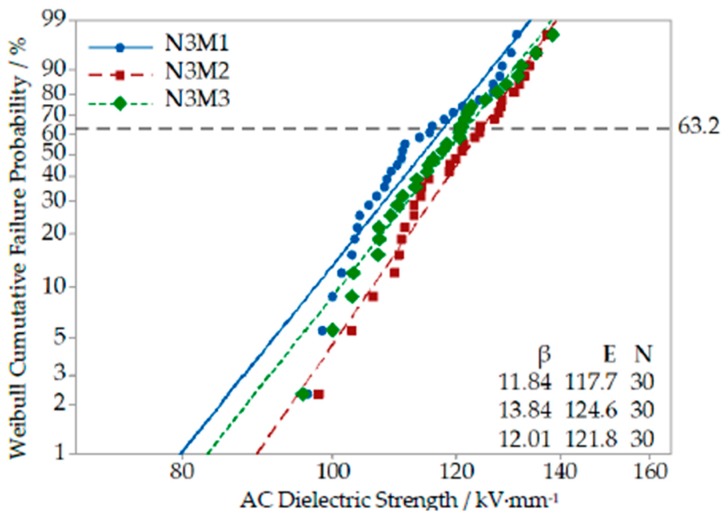
Weibull diagram of AC breakdown field strength of Micro-Nano-composites.

**Figure 10 polymers-12-00563-f010:**
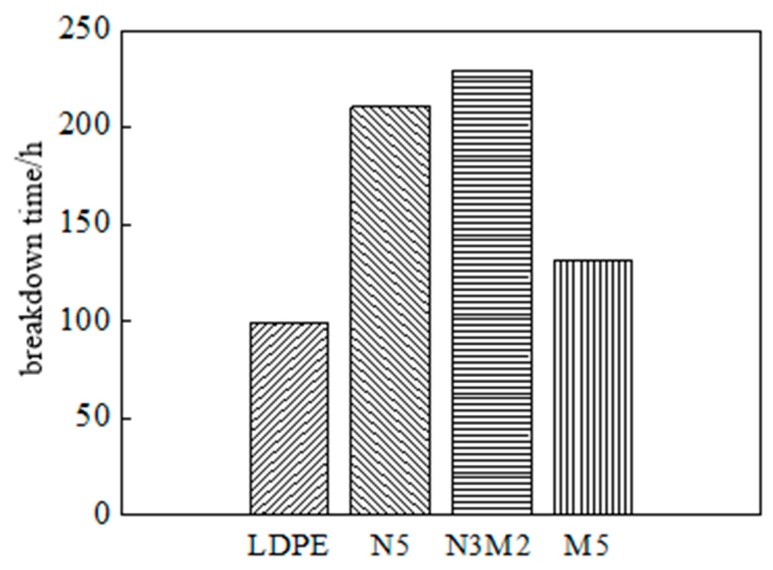
Breakdown time of corona for LDPE and ZnO/LDPE samples.

**Figure 11 polymers-12-00563-f011:**
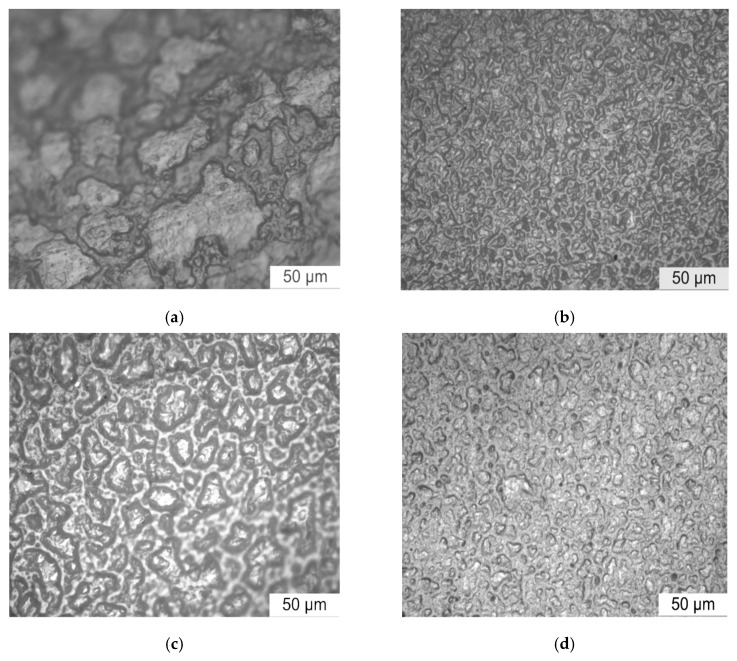
Corona corrosion of (**a**) LDPE; (**b**) N5; (**c**) M5; (**d**) N3M2.

**Figure 12 polymers-12-00563-f012:**
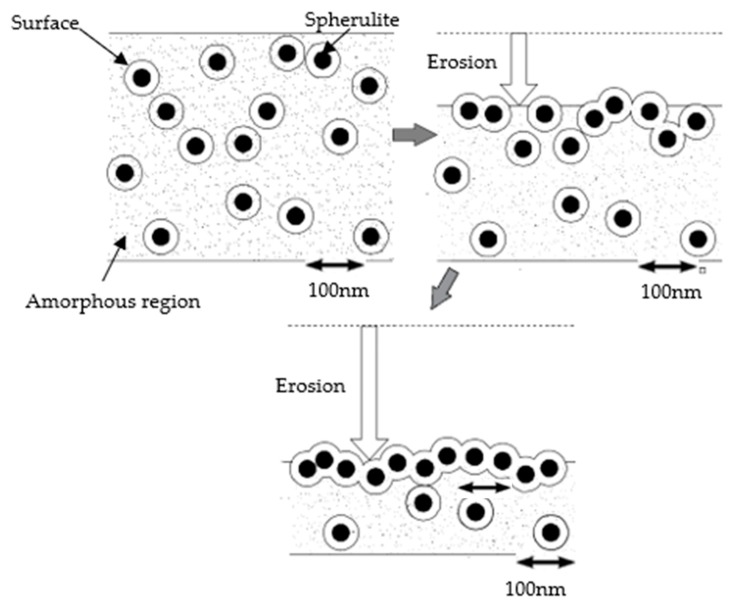
Corona corrosion process of polymer composites.

**Figure 13 polymers-12-00563-f013:**
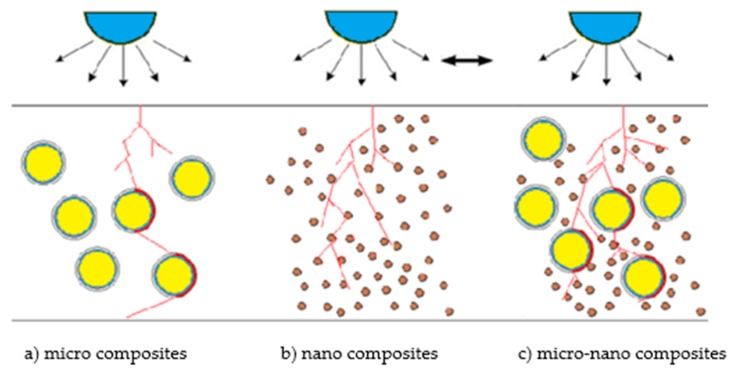
Electronics energy transport patterns of different composites.

**Table 1 polymers-12-00563-t001:** Isothermal crystallization and melting process parameters of micro, nano and micro-nano-ZnO/LDPE and LDPE specimens.

Sample	*T*_c_ (°C)	Δ*T*_c_ (°C)	*T*_m_ (°C)	*X_c_* (%)
LDPE	93.67	8.64	109.75	34.90
N1	94.38	6.22	110.04	36.65
N3	94.40	8.16	110.25	37.23
N5	95.15	7.50	109.84	36.18
M1	94.21	6.09	110.06	35.06
M3	95.12	6.02	110.60	35.64
M5	95.91	5.03	113.66	36.01
N3M1	93.96	7.90	109.98	36.58
N3M2	95.89	7.08	110.46	37.79
N3M3	94.64	7.82	111.32	35.94

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
