# Peer review of "The Research of Interface Microdomain and Corona-Resistance Characteristics of Micro-Nano-ZnO/LDPE"

_polymers, 2020, doi:10.3390/polym12030563_

Round 1

Reviewer 1 Report

In this article, LDPE composites with ZnO nanoparticles and microparticles were prepared and studied, investigating their interface microdomain and corona-resistance properties. Major modifications are required for its publication:

1) The authors have to highlight the novelty of their work.

2) Three XRD patterns are shown in Fig. 4. It is not clear to which sample each pattern corresponds, specifically for the case of raw ZnO and ZnO/LDPE. Also, it is convenient to include the pattern (in bars) of the standard card JCPDS72239.

3) Authors associate the increment of the diffraction peak intensities and the peak narrowing after modification to the bonding interaction formed between Nano-ZnO particles and silane coupling agents. Could it be due to the coalescence of the particles?.

4) What is the crystal size obtained from the XRD patterns?.

5) The peaks of the XRD pattern in blue in Fig. 4 are shifted with respect to the peaks of the other two patterns. It is not explained in the text.

6) Authors claim that particles with different particle size dope into pure LDPE, however, the particle size is not characterized.

Reviewer 2 Report

In this study, the authors prepared Micro-ZnO/LDPE, Nano-ZnO/LDPE and Micro-Nano-ZnO/LDPE to improve the electrical insulation properties of polymer composites. Obtained research outcomes can be published after a major revision process. 

1) The manuscript should be obviously checked by a native speaker. Also, numerous typos were found.

2) Experimental method. The authors should provide information on whether the ZnO particles were synthesized or purchased. In both cases, corresponding information should be added. 

3) Provide the digital micrographs of used ZnO particles. In particular, shape and size are quite important factors. Information on size distribution should be added. 

4) XRD data. Usually, a broad peak due to the organic phase is expected at lower angles in the case of ZnO/LDPE samples. XRD plot should be started from 10 or 15 (2-theta) degrees. Please label figures in a better way, it is not clear which color representing which sample. 

5) Provide FTIR study of prepared samples to confirm the dispersion of ZnO in LDPE 

6) It is not clear why 5% mass fraction was tested only. It is highly suggested to test several more concentrations (of the same particle size).   

Round 2

Reviewer 1 Report

Authors have significantly improved their work. I recommend the publication of the amended version as it is.

Reviewer 2 Report

no more comments